# Recent Advances in Studies of Genomic DNA Methylation and Its Involvement in Regulating Drought Stress Response in Crops

**DOI:** 10.3390/plants13101400

**Published:** 2024-05-17

**Authors:** Youfang Fan, Chao Sun, Kan Yan, Pengcheng Li, Ingo Hein, Eleanor M. Gilroy, Philip Kear, Zhenzhen Bi, Panfeng Yao, Zhen Liu, Yuhui Liu, Jiangping Bai

**Affiliations:** 1State Key Laboratory of Aridland Crop Science, College of Agronomy, Gansu Agricultural University, Lanzhou 730070, China; fanyf@st.gsau.edu.cn (Y.F.); lipengcheng@st.gsau.edu.cn (P.L.); bizz@gsau.edu.cn (Z.B.); yaopf@gsau.edu.cn (P.Y.); liuzhen@gsau.edu.cn (Z.L.); lyhui@gsau.edu.cn (Y.L.); 2School of Biological and Pharmaceutical Engineering, Lanzhou Jiaotong University, Lanzhou 730070, China; yank@mail.lzjtu.cn; 3The James Hutton Institute, Dundee DD2 5DA, UK; ingo.hein@hutton.ac.uk (I.H.); elea-nor.gilroy@hutton.ac.uk (E.M.G.); 4International Potato Center (CIP), CIP China Center for Asia Pacific (CCCAP), Beijing 102199, China; p.kear@cgiar.org

**Keywords:** plant, epigenetics, DNA methylation, drought stress, mechanism, research advances

## Abstract

As global arid conditions worsen and groundwater resources diminish, drought stress has emerged as a critical impediment to plant growth and development globally, notably causing declines in crop yields and even the extinction of certain cultivated species. Numerous studies on drought resistance have demonstrated that DNA methylation dynamically interacts with plant responses to drought stress by modulating gene expression and developmental processes. However, the precise mechanisms underlying these interactions remain elusive. This article consolidates the latest research on the role of DNA methylation in plant responses to drought stress across various species, focusing on methods of methylation detection, mechanisms of methylation pattern alteration (including DNA de novo methylation, DNA maintenance methylation, and DNA demethylation), and overall responses to drought conditions. While many studies have observed significant shifts in genome-wide or gene promoter methylation levels in drought-stressed plants, the identification of specific genes and pathways involved remains limited. This review aims to furnish a reference for detailed research into plant responses to drought stress through epigenetic approaches, striving to identify drought resistance genes regulated by DNA methylation, specific signaling pathways, and their molecular mechanisms of action.

## 1. Introduction

DNA structure in plants, like other organisms, consists of a double helix formed by nucleotide sequences. These nucleotides contain the bases adenine, thymine, cytosine, and guanine, which pair specifically (A with T and C with G) to form the genetic code. Plant DNA encodes the instructions for building and maintaining the plant, including the development of various traits and the regulation of metabolic processes [1]. Modifications of DNA methylation affect gene activity, which in turn affects plant growth and development and adaptation to adversity; throughout the process, these modifications regulate gene activity without altering the DNA nucleotide sequence. Thus, epigenetics has become a popular area of study in recent years. This allows genes to change their function and expression, which can then be passed down to offspring in a stable manner in response to environmental changes [2]. This preserves organisms’ normal growth and development and enables them to adapt to stress brought on by unstable environmental factors. Epigenetic modifications include DNA methylation, histone modifications, chromatin remodeling, and non-coding RNA (ncRNA) etc., which affect the structure and accessibility of chromatin, thus dynamically regulating gene expression [1].

DNA methylation has become one of the most thoroughly investigated research fields among categories of epigenetic modification, with the rapid advancement in high-throughput sequencing technologies for genome DNA methylation. As a conserved epigenetic modification, DNA methylation is a widespread covalent modification in biological genomes, which can regulate the corresponding functions of genomes without changing the primary structure of DNA molecules. It plays a vital role in gene regulation and genome stability [3]. DNA methylation is a form of chemical modification of DNA, which occurs by the transfer and covalent binding of methyl groups (CH3-) from S-adenosylmethionine (SAM) to sites on the DNA sequence where methylation can occur, such as adenine N-6, guanine G-7, and cytosine C-5 [4]. The process is catalyzed by various DNA methyltransferases (DNMT) after DNA replication. The DNA is further modified by DNA methylation to form N6-methyl purine (N6-mA), 7-methyl guanine (7-mG), and 5-methyl cytosine (5-mC) [5,6]. Among them, 5-mC occurs most frequently in eukaryotic organisms in the methylation site enriched region and has a role in regulating the transcription of biological genes [7]. It also allows organisms to respond to the environmental changes by maintaining genome stability, regulating gene expression, and altering genetic phenotype without altering DNA sequences.

The United Nations Intergovernmental Panel on Climate Change (IPCC) reports that the average temperature will rise by 1.8–4.0 °C by 2100 and that many regions of the world will face environmental change problems due to increased drought. Under the circumstances of external water stress coupled with lower water tables, drought has become a major threat to crop cultivation worldwide [8]. The impact of drought has been exacerbated by climate change in recent years, as drought stress has led to stunted crop growth, devastating effects on crops, and severe yield losses [9]. Drought is a significant contributor to agricultural production loss, particularly affecting least developed countries (LDCs) and low to middle income countries (LMICs). It has been identified that over 34% of crop and livestock production losses in these regions are attributed to drought, resulting in an economic impact estimated at approximately USD 37 billion. Notably, the agricultural sector bears the brunt of drought impacts, accounting for 82% of all such impacts. This underscores the profound effect of drought on food production and economic stability (FAO: http://www.fao.org/ (accessed on 7 April 2024). In recent years, DNA methylation has often been found in plants in response to biotic and abiotic stresses. For example, in biotic stress, the total DNA methylation level of rice was decreased in response to bacterial infection [10]; in abiotic stress, there was an increased genome-wide level of methylation in rice under high salt stress [11]; under heat stress, different levels of heat stress induced different levels of DNA methylation in *Arabidopsis thaliana* [12]; cold stress caused rice’s demethylation of the promoter region of cold tolerance genes [13]; and UV stress induced demethylation of the promoter region of key factors in Arabidopsis [14], leading to differential DNA methylation. In response to different patterns of climate change, drought stress caused an increase in the overall methylation level of linseed [15]. One study of cotton (*Gossypium hirsutum* L.) was a two-year field trial to assess cotton yield and its stability, linking DNA methylation patterns to plant development and yield under field conditions [16]. Our goal in the current review is to examine research progress regarding DNA methylation responses to drought stress to identify potential opportunities for the improvement of crop breeding for drought resistance.

## 2. Characteristics of DNA Methylation Modifications

In 1925, 5-mC was first discovered in the hydrolysis product of tuberculinic acid from the nucleic acid of the tubercle bacillus [17]. In subsequent studies, higher levels of DNA methylation modifications of 5-mC were also found in plants [18].

DNA methylation is based on the nature of the target gene and its function is reflected in the regulation of gene expression, transposon silencing, chromosomal interactions, and genetic characteristics [4].

DNA methylation is also crucial for maintaining genomic stability [19]. DNA methylation in plants can maintain genomic stability by inhibiting transposon and exogenous gene transcription to reduce genome disruptions such as translocation and recombination. Recent studies have revealed that transposons show high methylation levels after amplification, which can indicate that transposons in a highly active state can be regulated by DNA methylation and further act as a repressor. 

DNA methylation modification of genes in general is often found to exhibit a negative correlation with gene transcription [20,21], i.e., the higher the frequency and level of DNA methylation, the lower the level of gene transcription, which in turn affects the expression of the corresponding gene, this further increases the phenotypic differences between organisms. This is the reason why the same organism with the same whole genome sequence has different traits that are adapted to environmental changes [22]. This is in contrast to the methylation that occurs on transposons, where CG methylation within the gene region allows for moderate expression of the corresponding gene, mostly at different tissue sites and does not silence the gene [23,24].

In most eukaryotes, the most frequent DNA methylation usually occurs in the cytosine base, which contains CG, CHG, and CHH (H for A, T, or C, respectively) [20]. Among these three types, the CG methylation level is significantly higher than the CHG or CHH methylation types [25]. CG methylation can occur in the promoter region and part of the pre-transcriptional region, as well as in the 3′ end and part of the post-transcriptional region and may inhibit gene expression. There are two theories that explain this inhibition of expression. First, methylation that occurs in the promoter and enhancer regions prevents the binding of transcription factors required for gene transcription, affecting gene transcription and thereby inhibiting or even preventing gene expression; Second, cytosine sites that have undergone methylation can attract proteins bound to them, causing histone deacetylases and chromatin remodeling proteins to be attracted and chromatin to be compressed. This structural change results in the inability to transcribe, thereby inhibiting gene expression [21]. Although most DNA methylations occur in promotor regions and inhibit gene transcription, such methylation also acts as a promoter in a small number of cases [3].

Compared with mammals, the proportion of 5-mC in plant genomes is relatively high, and the genome-wide cytosine methylation level varies from species to species, ranging from 6 to 25% in different species [26]. For example, in the model plant Arabidopsis, 5% of cytosine is methylated [27]; in contrast, 24.3% of cytosine is methylated in young spikes of wild and cultivated rice, and in wheat it is over 20%. This significant difference is caused by the enrichment of repetitive sequences. The level of cytosine methylation also varies significantly in different regions of the same genome, and several studies have demonstrated that cytosine methylation levels are tissue-, organ-, and developmental stage-specific [20]. DNA methylation of transposable elements (TEs) is strikingly similar across species and DNA repeat sequences, with 50% methylation differences between ecotypes [4]. Methylation polymorphisms were found to occur most frequently in the upstream or downstream regions of genes after repressing the transcription of related genes, making its level negatively correlated with gene expression levels. Although the effect is not significant, studies on plants have mostly shown that DNA methylation also occurs in the gene body. However, it has been discovered in a study of poplar that methylation in the gene body significantly inhibits gene transcription more than methylation in the promoter region [28].

The first genome-wide methylation map of plants was published in 2016 by Zhang et al. [29] using the model plant *Arabidopsis thaliana* as a guide. The map’s findings revealed that more than one-third of the expressed genes in the genome were methylated. There is a significant difference in that only 5% of the genes are methylated in the promoter region. Based on this genome-wide methylation map, it was found that Arabidopsis genes that were methylated in the gene region were expressed and could reach high expression levels. Subsequently, Inagaki et al. [30] found that gene region methylation was higher in transcribed regions than in non-transcribed regions; however, the mechanism of this association is unclear. Graaf et al. [31] evaluated the mutation rate per CpG site per haploid per generation in Arabidopsis and found that the forward mutation rate (i.e., methylation gain rate) was about 2.56 × 10^−4^ and the reverse mutation rate (i.e., methylation loss rate) was about 6.30 × 10^−4^ in Arabidopsis, and these methylation mutation rates are about five times higher than the mutation rates found by Ossowski et al. [32].

## 3. Methylation Detection Methods

DNA methylation detection technologies are significant in epigenetics research, offering insights into gene regulation and its role in development, disease, and environmental responses. These methods enable the identification of methylation patterns across the genome, elucidating how epigenetic modifications influence gene expression and contribute to biological diversity and complexity. Moreover, knowing the dynamics of DNA methylation helps in exploring the developmental processes, environmental adaptability, and evolutionary mechanisms of organisms to improve stress resistance and productivity of crop plants [33]. Thus, the development and refinement of DNA methylation detection technologies continues to be a cornerstone of epigenetic research, with broad implications for biology and medicine. Therefore, by conducting a thorough search using PubMed and Google Scholar, these methods were identified as techniques for detecting DNA methylation, each offering different advantages, such as coverage, resolution, and cost.

### 3.1. Methylation Sensitive Amplified Polymorphism (MSAP)

MSAP is based on the amplified fragment length polymorphism (AFLP) method [34]. Methylation-specific isoschizomer HpaII and MSPI with a restriction endonuclease and genomic DNA, instead of AFLP functional enzymes and target bands, double digestion in order to obtain DNA fragments of different sizes, then join the enzymatically cleaved DNA fragments with the corresponding restriction endonuclease as a junction, then design the primers according to the junction. Although both enzymes can recognize the same site and have different methylation sensitivity, the amplified bands are different, based on which the methylation level of gDNA can be detected. Due to the high methylation specificity stemming from the coordinated activity of two methyltransferase enzymes, gDNA methylation levels can be further refined to distinguish between holo- and hemi-methylation states [35]. This assay is mostly used in early DNA methylation studies.

### 3.2. High Performance Liquid Chromatography (HPLC)

HPLC is divided into normal-phase HPLC and reversed-phase HPLC, which, in the study of calf thymus and salmon sperm, Kuo et al. [36] first detected DNA methylation using reversed-phase HPLC (RP-HPLC) [37], suggesting that HPLC can be a reliable method for detecting gDNA methylation levels. The DNA methylation peaks were obtained by RP-HPLC using the products of hydrolysis by specific deoxyribonucleases, nucleases, and bacterial alkaline phosphatases, and the DNA methylation levels were detected by further calculating the 5-mC content of gDNA and its ratio to cytosine. On this basis, the deformed HPLC was linked with PCR to form the DHPLC-PCR method system by Baumer, and this method makes DNA methylation detection more convenient and efficient [38]. As the research progressed, it was updated to high performance liquid chromatography-mass spectrometry (HPLC-MS) [39], which further improved the method technique for 5-mC detection.

### 3.3. Methylated DNA Immunoprecipitation-Sequencing (MeDIP-Seq)

MeDIP is a technique that uses monoclonal antibodies or DNA methylation-binding proteins that bind specifically to methylation sites to quantitatively capture enriched methylated DNA against 5-mC in the sample [40]. Highly methylated regions of gDNA can be identified, but not at the level of single base methylation.

### 3.4. Amplified Fragment Single Nucleotide Polymorphism and Methylation (AFSM)

The AFSM test employs restriction endonucleases with varying sensitivity to methylation to double cleave the genome and produce DNA fragments of various sizes for the effective detection of DNA methylation [41]. This assay is based on the lower cost and higher accuracy of second-generation sequencing technology [42,43]. AFSM is currently the only method in the world that can simultaneously detect single-nucleotide polymorphisms (SNPs) by high-throughput, insertion and deletion (inDels), and methylation sites in the whole genome with high throughput.

### 3.5. Methylation Sensitive Restriction Endonuclease (MSREs)

MSREs are a class of restriction endonucleases that are methylation-sensitive at the recognition site [44]. The fragment is obtained by cleaving the CpG sequence using its isozyme, which is insensitive to methylation, and then analyzed by Southern Blot. The MSREs method is a method that combines the sensitivity of methylation and the specificity of restriction enzymes to identify the methylation status of CpG sequences. It is convenient because it does not require detailed information about the sequence of the entire gDNA and the primary structure of DNA, but its application is more restricted because it needs a great deal of DNA with a high relative molecular mass and can only detect methylated alleles with a high copy number ratio.

### 3.6. Bisulfite Sequencing PCR (BSP)

BSP was first proposed by Frommer et al. [45] to be applied to 5-mC detection, where gDNA was first treated with hydrosulfite to react unmethylated cytosine C into uridine U [46]. The PCR reaction was carried out by specific primers to convert uridine U to thymine A, which was combined with high-throughput sequencing technology to distinguish 5-mC from other bases. In a subsequent study by Bianchessi et al. [47] to detect methylation in the mitochondrial DNA, non-coding region of endothelial cells, it was found that 5-mC was not randomly scattered but aggregated within the DNA coding region. Many other studies have shown that the BSP is still the most often used assay because it is accurate and dependable and can identify the methylation status of individual CpG sites despite the BSP’s limitations in detecting DNA methylation, such as the high cost and complexity of the method [48].

### 3.7. High-Performance Capillary Electrophoresis (HPCE)

It is a kind of product separation using the principle that narrow pore fused silica capillaries [49]. It can separate different chemical components from the complex to achieve quantitative detection of modified DNA by using the different charge properties, structure size, and chemical properties of DNA hydrolysis products in the background of the strong electric field. The main drawbacks of using HPCE for DNA methylation detection are high costs, complex sample preparation, technical demands, and dependency on chemical modification stability, potentially affecting result accuracy.

### 3.8. TET Enzyme-Assisted Pyridineborane Sequencing (TAPS) and Enzymatic Methyl-Seq (EM-Seq)

Both techniques use enzymatic and chemical reactions to complete sequencing to prevent degradation by bisulfite stimulation of most DNA. TAPS uses TET1 oxidase to oxidize 5-mC and 5-hmC to 5-caC, which is chemically converted to DHU by the reducing agent pyridine borane, which is then used as a template to be recognized by the corresponding DNA polymerase for U base, producing a C to T conversion by PCR amplification products [50]. Similarly, EM-seq uses TET2 and oxidation enhancer to oxidize 5-mC and 5-hmC to 5-caC, and then deaminates cytosine with APOBEC3A to deaminate the unmodified C to U, which is then recognized [51]. TAPS and EM-seq both offer precise DNA methylation data but face challenges. TAPS is technically complex and costly, requiring specialized data analysis. EM-seq is also expensive and technically demanding, which may restrict its use. Cost and technical feasibility are key considerations for both methods.

### 3.9. Reduced Representation Bisulfite Sequencing (RRBS)

Reduced Representation Bisulfite Sequencing (RRBS) represents an economical approach for analyzing DNA methylation. This technique utilizes restriction enzymes, such as MspI, to selectively digest the genome, thereby enriching for fragments from regions with high CpG content [52]. Following size selection, these fragments are treated with bisulfite, which converts unmethylated cytosines to uracil while preserving methylated cytosines. The fragments are then sequenced using high-throughput methods to assess methylation status. The primary advantages of RRBS are its cost-effectiveness and the targeted analysis it provides of critical gene regulatory regions. However, the technique does face challenges with coverage limitations and potential biases in fragment selection. RRBS is particularly valuable for focused studies on methylation in CpG-rich areas, making it indispensable for research into gene expression regulation.

### 3.10. Methylation Capture Sequencing (MCS)

Methylation Capture Sequencing is a sophisticated technique for DNA methylation analysis that leverages the specific affinity of either Methylated DNA Immunoprecipitation (MeDIP) or Methyl-CpG Binding Domain (MBD) proteins to selectively enrich methylated DNA fragments [53,54]. Once enriched, these fragments are purified, amplified, and organized into libraries that are optimized for sequencing. This method targets regions of high methylation with enhanced sensitivity, offering a cost-effective alternative to whole-genome sequencing. Despite its advantages, Methylation Capture Sequencing presents certain limitations, including restricted coverage and data accuracy that depend critically on the specificity and affinity of the utilized antibodies or proteins. Overall, this technique is cost-efficient and versatile, proving especially effective for research projects focused on specific methylation regions.

## 4. Mechanisms of Methylation Change Patterns

### 4.1. Mechanism of Methylation Action

DNA methylation is closely related to genome maintenance, parental imprint formation, and transcriptional regulation, and it is important to clarify the molecular mechanism for further research. It is known that the dynamic changes of DNA methylation includes three processes: de novo methylation, maintenance methylation, and demethylation.

#### 4.1.1. De Novo Methylation

Although CG and CHG methylation can occur via de novo and maintenance methylation pathways, the asymmetric CHH methylation type exclusively relies on the de novo methylation pathway. Following replication completion, unmethylated cytosine undergoes methylation through the activity of the corresponding methyltransferase. De novo methylation is mediated in plants by RNA, i.e., there is small interfering RNA (siRNA), scaffold RNA, and the corresponding protein DNA methylation pathway [55]; the de novo methylation can be divided into the classical RNA-directed DNA methylation (RdDM) pathway (Figure 1) and the non-classical RdDM pathway. As shown in Figure 1, classical RdDM is divided into two major steps. The first step is the synthesis of siRNA precursor RNA, i.e., Pol IV-dependent RNA (P4RNA). The SAWADEE HOMEODOMAIN HOMOLOG 1 (SHH1) protein binds to the H3K9 histone modified by methylation of the lysine at the ninth position of the tail through the Tudor-like fold structure of the SAWADEE structural domain, and then recruits RNA polymerase Pol IV to the specific site to synthesize single strand RNA (ssRNA) [56,57]. It was found that in Arabidopsis, this ssRNA, called P4RNA, is a precursor of 24 nt siRNA. ssRNA is synthesized into double-stranded RNA (dsRNA) by RNA-dependent RNA polymerase 2 (RDR2) [56,57]. It is further cleaved by the cleavage-like enzyme Dicer-like protein 3 (DCL3) into 24 nt siRNA [58,59], which requires the action of RNA methyltransferase HUA ENHANCER 1 (HEN1) to prevent degradation by other nucleases and maintain stability [60,61]. The mature 24 nt siRNA is loaded onto the effector protein Argonaute 4 (AGO4), mainly AGO4, and degrades the new strand generated by the action of RDR2 for pairing. The second step is transcription to produce scaffold RNA. While the stable 24 nt siRNA is loaded onto AGO4, the scaffold RNA is transcribed from the DNA damage repair (DDR) protein complex (DRD1/DMS3/RDM1), which interacts with the suppressor of Variegation Homologous2/9 (SUVH2/9) protein to attract Pol V to the specific site [62,63]. After complementary pairing of siRNA bases [64], it recruits domain rearranged methylase 2 (DRM2) to complete the de novo methylation through protein catalysis of multiple RdDM pathways. The main difference between non-classical RdDM and classical RdDM is the small RNAs (sRNAs) that mediate methylation [65], i.e., sRNAs other than the 24-nt heterochromatic siRNA (hetsiRNA) can also mediate DNA methylation to occur [65,66,67,68,69]. In addition, a few scaffold RNAs can also be obtained by Pol II transcription [70]; AGOs other than AGO4 have also been partially found to mediate DNA methylation [71,72].

#### 4.1.2. Maintenance of Methylation

Maintenance methylation is performed based on semi-conserved replication, i.e., when the parental chain originally contains a methylation site, the methylation modification occurs on the new synthetic chain paired with it. By this semi-conserved replication, maintenance methylation allows methylation to occur at two symmetric sites, CG and CHG, but maintenance of methylation at CHH asymmetric sites can only occur by de novo methylation [73]. Further, CG-type methylation is thought to be maintained by a simple replication mechanism that allows methylation to occur [55], whereas CHG-type methylation is more complex and requires maintenance of methylation through a combination of H3K9-containing and SRA-containing proteins [55,74,75]. However, the two are not independent of each other; rather, they influence each other. For example, CG methylation can act to maintain CHG methylation, while the specific site of CHG methylation can determine CG methylation [76]. The maintenance of DNA methylation in plants is associated with cytosine sequences and is regulated by different mechanisms catalyzed by DNA methylation transferases (Figure 2). One of the first homologous mammalian proteins catalyzing CG site methylation in plants, methyltransferase 1 (MET1), was identified (Figure 2A) [77]. Chromomethylase (CMT)-specific transferases that maintain CHG methylation are endemic in plants [78]. Specifically, it was demonstrated that the chromethylase3 (CMT3) with or without the loss of SUVH4 influenced whether DNA methylation levels were significantly lower or higher [79]. This suggests that CMT3 and SUVH4 are associated with CHG methylation. The histone methyltransferase SUVH4 structural domain binds to the hemimethylated site through its SRA structure, causing histone H3K9 in this site region to undergo methylation to produce H3K9me2, thereby recruiting CMT3 to interact with the specific site, causing CMT3 to bind to nucleosomes, and hemi methylation to become fully methylated and maintain the original DNA methylation [62,80] (Figure 2B). It has been shown that the active state of the chromatin remodeling factor decreased and that methylation 1 (DDM1) is important for the maintenance of MET1 and CMT3, two methylation transferases [81]. DRM2 and CMT2 maintain the methylation of asymmetric CHH through different pathways, with DRM2 maintaining the methylation status of the RdDM target region through the de novo methylation pathway (Figure 2C) [82] while CMT2 catalyzes the methylation of CHH containing histone H1 and heterochromatin [80]. In addition, it has also been shown that MET1 and CMT3 are involved in the maintenance of asymmetrically methylated CHH [63,83]. However, overall, the RdDM pathway plays a crucial role in the maintenance of CHH methylation.

#### 4.1.3. Demethylation

DNA methylation can inhibit biological gene regulation, and there must be a mechanism in the organism that dynamically balances with methylation to maintain a stable state. This mechanism is demethylation. Demethylation refers to the change of reverting a site originally modified with methylation to cytosine. It may be classified into two types based on the mechanism: passive demethylation and active demethylation [84].

##### Passive Demethylation

Passive demethylation inhibits the maintenance of de novo methylation and symmetric site methylation [85]. After passive DNA demethylation acts on DNA replication, when the DNA strand with methylation modification is replicated semi-conservatively, the methylation-dependent DNA transferase activity decreases or the concentration does not reach the required level, and the corresponding site where the methylation modification occurs is still cytosine, resulting in the loss of methylation of the newly synthesized strand [86]. The final level of DNA methylation in the organism is reduced [3]. However, this passive demethylation, based on semi-conserved replication, is far from meeting the need to inhibit DNA methylation and prevent gene silencing, so active demethylation is still the main way to cope with environmental changes.

##### Active Demethylation

Active demethylation is a specific enzymatic reaction involving DNA glycosylases and cleavage enzymes such as repressor of silencing1 (ROS1), demeter (DME), demeter-like2 (DML2), and demeter-like3 (DML3) enzymes [87,88]. In this process, the E3 ligase enhances the stability of ROS1 to advance the reaction [89], recognizes the 5-mC at the site of DNA methylation, and then hydrolyzes it to break the glycosidic bond and remove the methylated cytosine from the DNA backbone [90]. In combination with the base excision repair (BER) mechanism, the synthesized unmethylated cytosine is used to complete the repair by completing the gap to achieve active demethylation [86,91]. This is also consistent with Ikeda et al. [92] and Zhu’ s findings [93]. As shown in Figure 3, there are currently three different pathways to achieve active demethylation, the first of which is mediated by the first DNA active demethylation complex identified in plants, where the structural domain protein neuronal pentraxin 1 (NPX1) and the methyl-CpG-binding domain protein 9 (MBD9) can preferentially recognize acetylated histone marks established by increased DNA methylation 1 (IDM1) on CG-type methylation, thereby recruiting the INO80 chromatin remodeling complex SWR1 to specific chromatin to deposit variant H2A of histone H2A.Z, which further recruits the ROS1 to specific target sites to complete demethylation [94]. The second one is mediated by RWD40, the first DNA demethylation complex containing DNA demethylase ROS1 found in Arabidopsis. ROS1 recruits structural domain protein RWD40 to specific sites and interacts with DNA methylation binding protein RMB1, zinc finger, and structural domain protein RHD1 to form the RWD40 complex, which regulates endogenous site methylation of all methylation types. It can also regulate the gene expression level of ROS1 through the ROS1 gene promoter, thus completing the active DNA demethylation to regulate DNA methylation level [95]. Third, the AGENET domain-containing protein 3 (AGDP3), identified by forward genetic screening, recognizes and binds to the methylated histone H3K9me2 on the one hand, while on the other hand it recruits ROS1 to target the genomic site and causes the methylated site to be demethylated by base excision repair [96].

### 4.2. Pattern Variation and Genetic Characteristics

Factors that cause changes in DNA methylation are classified as endogenous and exogenous [97]. Exogenous factors are environmental changes that have a significant impact on an organism’s ability to grow and are categorized as biotic and abiotic stresses. Endogenous factors are methylation changes brought on by changes at the genetic level, such as transposon insertions and deletions, chromosome rearrangements, and mutations in methylation-related factors. However, it has been suggested that exogenous factors are more influential than endogenous factors for heritable DNA methylation changes from a long evolutionary perspective [98]. The level and status of DNA methylation is not fixed in an organism, and there is a distinction between “transient” and “long-lasting”. The majority of cases are “transient” and are dynamically regulated to meet the needs of the organism in response to changes in DNA methylation through the involvement of a series of related enzymes that target specific sites through different pathways [73]. The state and level of a dynamic balance of DNA methylation, which can be passed on through DNA replication as a relatively stable imprint of epistatic modifications, is generally in a relatively stable state in the organism, but can be changed when stimulated and passed on to offspring, called stress memory, which is a “long-lasting” situation [99]. Depending on the duration of the memory, it can be divided into short-term somatic memory, which is caused by physiology and metabolism for a few days or weeks, and long-term intergenerational stress memory, which is inherited through mitosis and meiosis [55,100].

In the study by Williams [101], *Arabidopsis thaliana* regulates 5-mC glycosylases through corresponding methylation genes as a way to respond to stress and to inherit this memory across generations, giving offspring the ability to stably retain the associated resistance.

## 5. Effect of DNA Methylation on Plant Response to Drought Stress

Plants are rooted to a single point and they cannot change position to escape environmental challenges. This makes drought, an abiotic stress, one of the major limiting factors for the growth, development, and production of most plants worldwide [102]. Drought stress can disrupt the relatively stable equilibrium built up in plants, causing disruptions at the molecular level, resulting in physiological disorders, and further hindering growth and development, affecting key indicators such as yield, and even causing plant death [73]. In contrast, plants can also protect themselves against external damage by re-establishing the regulatory mechanisms of cellular homeostasis. Studies have demonstrated that drought stress can induce significant changes in DNA methylation levels, activate related signaling pathways, alter the expression of corresponding drought genes, and affect plant growth and development [103,104]. Among the gene families responding to drought stress, methylation and demethylation were found in genes in a CG background [105]; therefore, DNA methylation is thought to be involved in the regulation of plant drought stress response, and most DNA methylation variants can be transmitted from generation to generation, altering the physiological and ecological changes of plant growth and development to adapt to the environment [3,4].

### 5.1. Effect of DNA Methylation on Plant Growth and Development and Stress Resistance

DNA methylation is prevalent in the whole genome of plants and acts in different tissue regions with different methylation patterns at different developmental periods, which in turn specifically regulates genes to be expressed, or suppressed, in specific developmental stages, thus ensuring cell differentiation and normal plant growth and development [73]. Candaele et al. found that DNA methylation transferase in maize leaves, after the cells undergo a gradual movement of spatial gradients, induces CG and CHG methylation of different backgrounds in the division, transition, extension, and maturation zones, suggesting that DNA methylation plays an important role in regulating the growth and development of maize leaves [106].

In recent studies, it has been found that the biological clock is closely associated with DNA methylation. The biological clock is a complex regulatory system developed by plants over a long period of evolution by which organisms can sense the changing temporal patterns of their surroundings and adapt to environmental changes to survive [107]. DNA methylation, however, can directly affect the biological clock of plants and participate in the regulatory signaling network at the molecular level to meet and respond to the needs of plant growth, development, and stress [108]. It has been revealed in many studies that DNA methylation can regulate many key life activities of plants, such as flowering, immunity, maturation, etc., through the biological clock [109,110,111]. In tomato [112], strawberry [113], sweet orange [114], and pepper [115], it has been found that DNA methylation changes dynamically with the growth and development of the organism, and the corresponding changes in methylation levels can, in turn, act on the organism to promote or inhibit its maturation. It has also been found that inhibition of DNA methylation transferase can significantly prolong the biological clock in Arabidopsis [116].

Application of zebularine demethylation altered the development of drought-stressed parental polygonum seedlings, with significant changes in leaf area, root length, and biomass [117]; application of methylation inhibitor to spring wheat resulted in significant phenotypic changes, reduced malondialdehyde (MDA) content, increased proline and soluble sugar content, activated superoxide dismutase (SOD), peroxidase (POD), and catalase (CAT) enzyme activities under salt stress, and caused plant dwarfing, and significantly improved antioxidant capacity of wheat leaves under salt stress [118]. The exogenous application of the DNA methylation inhibitor 5-azadC to potato significantly inhibited the growth and development of potato, and the phenotype was significantly different from that of the control, with a significant decrease in plant dry weight, plant height, number of leaves, and total root length, and a significant increase in SOD and POD activity to alleviate the abiotic stress [119]. Meanwhile, transcriptomic studies revealed that genes of the MAPK signaling pathway, glutathione metabolism, glycolysis/gluconeogenesis, phosphatidylinositol metabolism, and phytohormone signaling pathways in potato plants responded to both drought stress and demethylation treatments [120].

### 5.2. Progress of DNA Methylation Involved in Drought Stress Response

Under abiotic stresses such as drought, DNA methylation regulates the expression of genes and activates, through dynamic changes in patterns and levels of gene expression, to cope with the damage caused by stress [4]. It has been shown that the polymorphic changes in DNA methylation sites induced by drought stress accounted for 12.1% of the genome-wide DNA methylation sites, thus affecting growth and development in response to drought stress, and 29% of the methylation was retained even after the subsequent stress was removed [121]. Therefore, DNA methylation is important for plant response to drought stress. Table 1 shows the current studies on DNA methylation changes in different plants species subjected to drought stress.

Based on our review of previous studies, we found that in general, plants change the extent of DNA methylation in response to drought stress. While two studies in Table 1 showed a genome-wide decrease in DNA methylation levels [122,125], and six studies showed an genome-wide increase in DNA methylation levels [124,128,136,139,140,142], an increase or decrease in methylation levels of the internal part of promoters of drought response genes was specifically determined by the positive or negative regulation of the gene under drought stress [126,127,130,134,135,137,140,143]. The proportion of methylation changes in response to drought mostly ranged from 10 to 20%, but some showed only slight changes, which may be related to tissue specificity [144]. In contrast, the proportions and trends of methylation sites in the three contexts were different, with the trends of symmetric methylation sites such as CG and CHG remaining the same, whereas most of the CHH asymmetric methylation sites were opposite to the symmetric methylation sites, which might be more sensitive to drought environment; this was confirmed in the study of *Zea mays* L. and *Populus tomentosa* [125,135], although methylation of CG, CHG symmetry sites were more sensitive to drought stress in other species of plants studied [126,143].

Recent studies, as outlined in Table 1, highlight the role of DNA methylation in plant responses to drought stress. These investigations build on foundational research, demonstrating significant changes in methylation patterns across various genomic regions when exposed to drought conditions. Plants appear to employ this dynamic epigenetic mechanism to adeptly manage environmental challenges. Importantly, these alterations in methylation predominantly impact genes and pathways crucial to the drought stress response, including water uptake and root architecture [135], osmotic balance [125], and hormone signaling processes [125,129]. Studies have shown that changes in DNA methylation patterns can activate or suppress the expression of specific genes involved in drought response. This regulatory modulation significantly affects the physiological and ecological attributes of plants, notably enhancing their drought resilience. We suggest that through such epigenetic modifications, plants precisely adapt their biological functions to better cope with drought conditions.

## 6. Conclusions and Outlook

Abiotic stresses such as drought have a huge impact on plants and can affect a wide range of plant growth and development, morphological indicators, and yield. Coupled with the fact that drought stress, an environmental problem, is a global issue, research related to drought tolerance and drought resistance in plants has been one of the key focuses of researchers. Epigenetics, a relatively new topical issue with respect to regulation of gene expression in response to environmental conditions, is also important as it can produce intergenerationally transmissible changes in phenotypic traits without altering the gene sequences.

The current review identifies efficient methylation detection technologies tailored to their specific objectives, facilitating further studies in this field. The current review has summarized the molecular mechanisms underlying the patterns and changes in DNA methylation, thus assisting researchers of drought responses to streamline and accelerate future efforts toward research progress.

Furthermore, we have summarized the details of the shifts in patterns, level changes, and physiological and ecological responses of DNA methylation to drought stress in Table 1. In our literature review, we identified numerous studies focusing on DNA methylation and drought stress, indicating significant interest in this area. However, most studies only demonstrated that plants exhibit DNA methylation changes at the whole genomic level under drought conditions. This review concentrated on studies that showed specified DNA methylation changes of pathways and genes, presenting data with defined changes in ratios between drought and control, which provides a solid foundation for more detailed investigations. These findings suggest that increased DNA methylation across the genome primarily serves to deactivate non-essential functions related to stress resistance. Changes in the methylation of promoter regions are tailored to the genes that respond to drought, influencing plant growth and development through various pathways, including the biological clock. These results in alterations to plant morphology and physiology, enabling adaptation to environmental shifts or survival under extreme conditions.

Efforts toward genetic improvement of crops using DNA methylation are still devoted to discovering loci that respond to drought stress, and verifying whether their methylation status is tightly correlated with plant drought resistance, then using advanced gene targeted editing technologies and biotechnological approaches [145,146,147], such as CRISPR/Cas9, to change the loci into a sustained methylated or demethylated state, so as to obtain new germplasm with enhanced drought resistance.

However, more specific studies, such as the molecular mechanisms of DNA methylation in response to drought stress, including the transmission of signaling molecules, the activation of related pathways, and the investigation of transcription factors, have not been adequately investigated; how DNA methylation occurs, and how it changes from a short-term genetic effect to a long-term genetic effect in response to environmental stress has not been clearly investigated [7]. Moreover, only a few studies of DNA methylation response to environmental stresses directly related to field trials. Looking ahead, we recommend that future research should focus on unraveling the complex interactions between DNA methylation and other epigenetic mechanisms in regulating plant stress responses. Detailed studies on the temporal and spatial patterns of methylation changes in response to varying stress levels could provide insights into the flexibility and resilience of plant epigenetic systems. Additionally, integrating epigenomic data with transcriptomic and metabolomic profiles would offer a holistic view of the plant’s response to stress, facilitating the identification of key regulatory nodes that can be targeted for crop improvement. In response to the global drought problem and food security challenges, the development of drought-resistant crop varieties by combining modern molecular breeding tools such as gene editing and molecular marker-assisted selection with epigenetic studies is of great significance for the breeding of drought-resistant varieties of plants and crops and germplasm innovation research, which can enrich the new strategies of plants to adapt to the adverse environments and solve the problem of food security during the earth’s climate change process [85].

## Figures and Tables

**Figure 1 plants-13-01400-f001:**
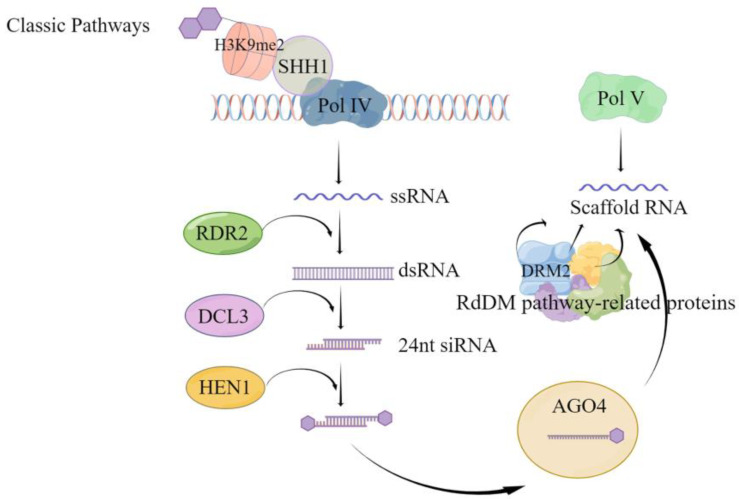
Classical RdDM pathway-mediated de novo methylation pattern. This figure illustrates the key steps in the classical RNA-directed DNA methylation (RdDM) pathway for de novo methylation. The process initiates with the recognition of target sites by the Pol IV complex associated with H3K9me2, facilitated by SHH1. The activity of the Pol IV complex produces single-stranded RNA (ssRNA) [56,57], which is then transcribed into double-stranded RNA (dsRNA) by RDR2. DCL3 processes the dsRNA to generate 24 nucleotide small interfering RNAs (24 nt siRNAs), which are subsequently methylated by HEN1 to enhance stability [61]. The siRNA is then guided to the target DNA sites in association with AGO4 [59]. With the assistance of RdDM pathway-related proteins, Pol V synthesizes scaffold RNA at the target site [65]. This scaffold RNA binds to the siRNA-AGO4 complex, directing the DNA methyltransferase DRM2 to the correct location and catalyzing the methylation process, culminating in de novo DNA methylation. The figure explicates the interactions between the components of the RdDM pathway and their roles in targeted DNA methylation.

**Figure 2 plants-13-01400-f002:**
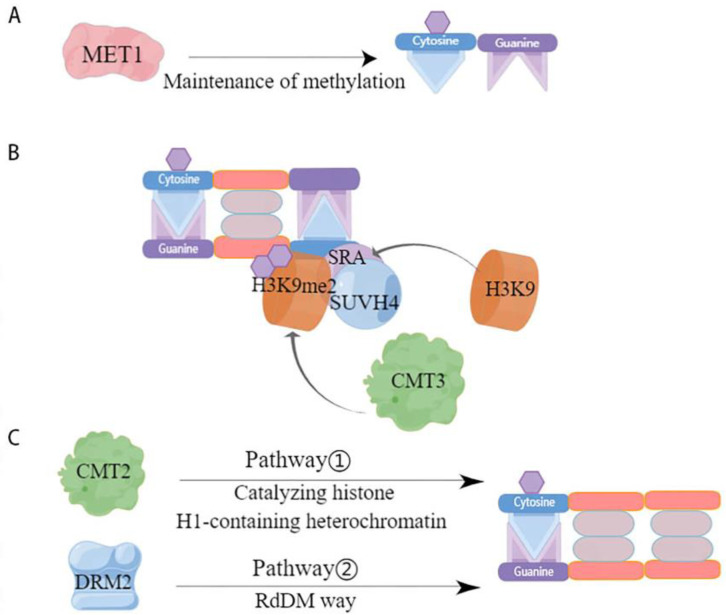
Mode of action of DNA methylation transferases involved in maintaining methylation. This figure depicts the various maintenance methylation pathways catalyzed by DNA methyltransferases. (**A**) CG maintains methylation. MET1 is a key maintenance methyltransferase for CG methylation [77]. It identifies the methylated cytosine on the parental DNA strand following replication and propagates the methylation pattern onto the daughter strand, ensuring consistent CG methylation across cell divisions; (**B**) CHG maintains methylation. SUVH4, through its SRA domain, recognizes H3K9me2, a marker of heterochromatin, and recruits CMT3 to methylate cytosine in the context of CHG. This is facilitated by a feedback loop where CMT3-mediated CHG methylation promotes H3K9 methylation by H3K9 methyltransferases, which in turn maintains the binding of SUVH4 and the recruitment of CMT3 [78]; (**C**) CHH maintains methylation. Two distinct pathways maintain CHH methylation. Pathway 1 involves CMT2, which methylates cytosine in the context of CHH in histone H1-containing heterochromatin regions, indicating a role beyond CHG methylation [81]. Pathway 2, the RdDM pathway, involves DRM2, which is guided by small interfering RNAs to target specific DNA sequences for CHH methylation, a process that is critical for silencing transposable elements and regulating gene expression.

**Figure 3 plants-13-01400-f003:**
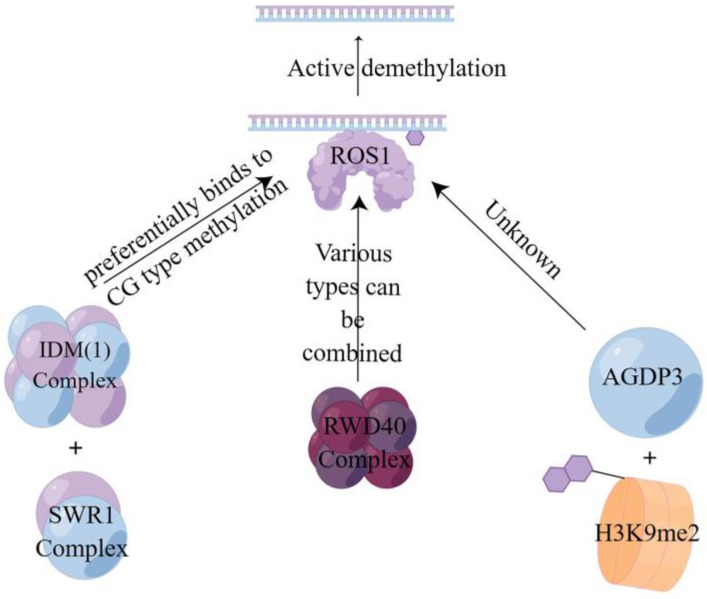
Active demethylation mode of action of three different pathways. This figure illustrates three pathways of the active DNA demethylation [94,95,96]. Three different active demethylation pathways all require the use of ROS1 to achieve them.

**Table 1 plants-13-01400-t001:** DNA methylation changes in different plants under drought stress.

Species	Processing	DNA Methylation Changes	Related Genes or Access	Associated Phenotypes	References
*Arabidopsis thaliana* (L.) *Heynh.*	After 7–30 dpg (days post germination) growth, stop water treatment for 20 days	A significant 15% decrease in the 5-meC content	Related to *DCL2*/*DCL3* pathway	Decreased homologous recombination frequency (Increased generally HFR, DNA hypermethylation, and higher stress tolerance)	[122]
	After 4 weeks of growth, the treatment group stopped water for 20 days	DNA methylation levels in the promoter region of AtGSTF14 were significantly reduced by 10%	*AtGSTF14*	N.A.	[123]
*Populus trichocarpa*	After 2 months of growth, the soil moisture content is controlled at about 10%	Significantly higher methylation levels of methylated cytosine, upstream 2 kp, downstream 2 kb and repetitive sequences (2–3% increase in the whole genome)	*C2C2*, *WRKY*, *MYB*, *EIL* gene family	N.A.	[124]
*Populus tomentosa*	After 2 months of growth, soil moisture content was controlled at 20–25% under 37 days	Significant reduction in genomic DNA methylation levels	*GATA9*, *LECRK-VIII.2*	Ceases leaf photosynthetic activity; Accumulation of ABA, osmolytes such as glycine betaine (BETA), proline (PRO) and osmotic regulator (ORS)	[125]
*Solanum * *lycopersicum*	Grow for 3 weeks to clean the roots and place on blotting paper under incandescent light until wilting occurs	Elevated CG methylation level in exon 1 of Asr1 and loss of methyl markers at CNN sites (mainly intron regions)	*Asr1*	N.A.	[126]
*Solanum pennellii*	Seedlings are removed from the soil and placed on filter paper	The DNA of the PKE1 promoter was highly methylated in fruit and leaf	*PKE1*	N.A.	[127]
*Oryza sativa*	Different tolerant cultivars	Elevated levels of genomic methylation in response to drought and salt	smRNA pathway	N.A.	[128]
	28 °C, air dry 80 min, rehydration 22 h after the cycle of treatment 2 rounds	DNA methylation regulates the expression of stress memory transcripts	ABA Access Road	Relative water content sharply dropped; the endogenous contents of ABA and JA phytohormones contents increased	[129]
	After 2 weeks of growth, treatment with 20% PEG6000 for 12 h	Genome and ZFP promoter and CDS region are highly methylated	*ZFP*	N.A.	[130]
	1/2 MS medium with 20% (*w*/*v*) PEG6000	JMJ710 demethylated H3K36me2 both in vivo and in vitro	*JMJ710*	The survival rates and water loss in the experiment with detached leaves are higher than check	[131]
*Zea mays* L.	Grown for 1 month, drought treatment for 9 days	Total methylation levels reduced around 20% in the maize ABA-deficient mutant vp10	ABA pathway	Leaf relative water content decreased rapidly	[132]
	Seedlings were not watered until they had three true leaves and were re-watered for six days when significant wilting was observed.	Sites nearest the MITE insertion, were hypermethylated in *ZmNAC111* promoter	*ZmNAC111*	Leaf photosynthesis rates (PS), stomatal conductance (SC) and transpiration rates (TR) were significantly smaller than check	[133]
	Grown for 1 month, drought treatment for 9 days	DNA methylation in the upstream region of the DBF1 gene	*DBF1*	The average relative water content was significantly higher than check	[134]
	Stop watering for 15 d when growth reaches the 5-leaf stage	DNA hypermethylation at CG and CHG sites and DNA hypermethylation at CHH site in the middle of ZmEXPB2 gene promoter (around 20% decrease)	*ZmEXPB2*	Significant decrease in fresh weight of whole plant and 6th leaf length, stunted secondary root growth, and increased primary root length	[135]
*Hordeum vulgare* L.	After germination, water deficit treatment for 10 d	High overall DNA methylation level	*HvDRM*	N.A.	[136]
	After 7 d of growth, stop hydroponics for 10 d	Methylation and demethylation of different regions of the *HvDME* promoter	*HvDME*	N.A.	[137]
*Solanum melongena* L.	After 3 weeks of growth, water was stopped for 2 d	Upregulation of demethylase expression	*SmelMET1*, *SmelCMT*, *SmelDRM*	N.A.	[138]
*Brassica juncea*	Watering was stopped for 15 d after seed germination until the leaves were yellow and curled.	Gene body methylation was increased in 90% of sites (around 10% decreased), while promoter methylation was gene function dependent	BAX inhibitor 1, metacaspase 4, B3, DIE2/ALG10, F-box, Bcl2	N.A.	[139]
*Morus alba*	Grown for 2 months (fresh leaves appear), 14 d water stop	0.5% Increased genomic DNA methylation	Phenylpropanoid biosynthesis and other multi-pathways	Relative water content (RWC) was decreased, leaf lengths were shorter	[140]
*Malus pumila Mill.*	Grown for 4 months, incubated with Hoagland solution containing 20% PEG8000 for 6 h (short-term) or 15 d (long-term)	Increased DNA methylation level of MdRFNR1-1 promoter	*MdRFNR1-1*	The fresh weights of all calli decreased; POD and CAT activities were lower in MdRFNR1 RNAi lines than in GL-3 plants	[141]
*Citrus unshiu Mac.*	Around 18–20% soil moisture content	High global DNA methylation level	*FLC*, *BFT*	A significant increase in the flowering branches, whereas an apparent decrease in vegetative branches	[142]
*Triticum aestivum* L.	Drought primed for 24 h via the addition of PEG 6000 at 10% (−0.36 MPa), 15% (−0.58 MPa) and 20% (−0.91 Mpa) for 72 h at the six-leaf stage	The CG and CHG methylation rates were decreased in *TaP5CS* and *TaBADH* promoters	*TaP5CS*, *TaBADH*	Plant dry weight and leaf area were significantly reduced, ΦPSII and increased ΦNPQ, higher photosynthetic rate and stomatal conductance	[143]

## Data Availability

Not applicable.

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
