# Peer review of "Recent Advances in Studies of Genomic DNA Methylation and Its Involvement in Regulating Drought Stress Response in Crops"

_plants, 2024, doi:10.3390/plants13101400_

Round 1

Reviewer 1 Report

Comments and Suggestions for Authors

The manuscript by Fan et al. reviewed recent studies on DNA methylation's role in plant drought stress responses, noting observed changes in methylation but a gap in identifying specific genes and pathways. It aimed to guide further research into drought resistance mechanisms. Overall, the manuscript describes some important findings that are of interest to a broad audience.

  1. How does DNA methylation in response to drought stress compare across different plant species? Are there common patterns or notable differences in how methylation regulates gene expression and development in response to drought?
  2. What are the main challenges hindering the identification of specific genes and pathways involved in the methylation response to drought stress? Are there particular strategies the authors recommend to address these challenges? 
  1. Considering the role of DNA methylation in drought stress response, what are the potential implications for developing drought-resistant crop varieties? Could the authors discuss any ongoing efforts or future research directions aimed at utilizing DNA methylation knowledge for crop improvement?
  2. Beneficial microbes are recognized for their role in enhancing host plants' responses to biotic and/or abiotic stresses, such as drought. Is there evidence to suggest that these beneficial microbes also play a role in the genomic DNA methylation of plants? The following papers provided related info on beneficial bacteria' roles in enhancing plant health against abiotic and biotic stressors and can also be cited:
    1. Yang, P., Liu, W., Yuan, P., Zhao, Z., Zhang, C., Opiyo, S. O., Adhikari, A., Zhao, L., Harsh, G., & Xia, Y. (2023). Plant Growth Promotion and Stress Tolerance Enhancement through Inoculation with Bacillus proteolyticus OSUB18. Biology, 12(12), Article 12. https://doi.org/10.3390/biology12121495
    2. Yang, P., Yuan, P., Liu, W., Zhao, Z., Bernier, M. C., Zhang, C., Adhikari, A., Opiyo, S. O., Zhao, L., Banks, F., & Xia, Y. (2024). Plant Growth Promotion and Plant Disease Suppression Induced by Bacillus amyloliquefaciens Strain GD4a. Plants, 13(5), Article 5. https://doi.org/10.3390/plants13050672
  1. For figures, references are missing.
Comments on the Quality of English Language
  1. Line 2, The -> the
  2. Line 3, Its  ->its
  3. Line 503, please revise accordingly in English.
  4. Line 724, please revise accordingly in English.
  5. Consider providing more detailed figure legends.
  6. Table 1, consider adding common name for plant species. E.g. Latin species name (common plant name) 

Reviewer 2 Report

Comments and Suggestions for Authors

Comments to authors:

The authors should add graphic abstract.

The authors would do English editing.

Introduction:  

1-      Could you please add a concise Text about DNA structure of the plant  to be more clear for readers?

2-      In lines 66-67, you should add recent statistics about Drought stress losses.

3-      What are the advantages and advantages of DNA methylation method compared to other methods for treatment of drought stress of plants.

4-      Are there field experiment of DNA methylation method on plants?

Methods

What are the tools used for data collection? please add the search engines.

Discussion

Could you add your own opinion and present your data in the light of literature survey?

 Conclusion

Conclusion text isn’t clear enough.

 What is the future perspective of this research? You can add your outcome of these data and your recommendations for other researches.

References:

References 41,31,15,21, 46; please try to cite recent ones.

Comments on the Quality of English Language

The authors would do English editing.

Round 2

Reviewer 1 Report

Comments and Suggestions for Authors

The authors have addressed the comments and made significant improvements to the manuscript. It can now be accepted pending a final check of grammar and English.

Comments on the Quality of English Language

NA

Author Response

Dear reviewer,

We sincerely appreciate your thorough review and constructive feedback on our manuscript. Your insights have been invaluable in refining our work, and we are grateful for the time and effort you have dedicated to evaluating our manuscript.

We are delighted to hear that you have acknowledged the efforts we've made in addressing your comments and implementing significant improvements to the manuscript. Your guidance has played a pivotal role in enhancing the clarity, coherence, and overall quality of our research.

Regarding your suggestion to conduct a final check of grammar and English, we fully agree with the importance of ensuring linguistic precision in our manuscript. We have already carried out a meticulous review focused on grammar and English usage, and we are committed to conducting another thorough examination to ensure that the manuscript meets the highest standards in this regard.

We are confident that the revised manuscript now meets the criteria for acceptance pending the final check of grammar and English. We are eager to provide any further revisions or clarifications as needed to facilitate the publication process.

Once again, we extend our gratitude for your valuable feedback and guidance throughout this review process. Your contributions have been instrumental in improving the quality of our manuscript.

Please do not hesitate to contact us if you require any additional information or have further suggestions. We are dedicated to addressing any remaining concerns promptly and efficiently.

Thank you for your continued support and consideration.

Best regards,

Yours sincerely,